# Effects of Combinations of Untranslated-Region Sequences on Translation of mRNA

**DOI:** 10.3390/biom13111677

**Published:** 2023-11-20

**Authors:** Anna Kirshina, Olga Vasileva, Dmitry Kunyk, Kristina Seregina, Albert Muslimov, Roman Ivanov, Vasiliy Reshetnikov

**Affiliations:** 1Translational Medicine Research Center, Sirius University of Science and Technology, 354340 Sochi, Russia; 2Laboratory of Gene Expression Regulation, Institute of Cytology and Genetics, Siberian Branch of Russian Academy of Sciences, 630090 Novosibirsk, Russia

**Keywords:** 5′UTR, 3′UTR, mRNA therapeutics, translation efficiency, RNA stability

## Abstract

mRNA-based therapeutics have been found to be a promising treatment strategy in immunotherapy, gene therapy, and cancer treatments. Effectiveness of mRNA therapeutics depends on the level and duration of a desired protein’s expression, which is determined by various *cis*- and *trans*-regulatory elements of the mRNA. Sequences of 5′ and 3′ untranslated regions (UTRs) are responsible for translational efficiency and stability of mRNA. An optimal combination of the regulatory sequences allows researchers to significantly increase the target protein’s expression. Using both literature data and previously obtained experimental data, we chose six sequences of 5′UTRs (adenoviral tripartite leader [TPL], HBB, rabbit β-globin [Rabb], H4C2, Moderna, and Neo2) and five sequences of 3′UTRs (mtRNR-EMCV, mtRNR-AES, mtRNR-mtRNR, BioNTech, and Moderna). By combining them, we constructed 30 in vitro transcribed RNAs encoding firefly luciferase with various combinations of 5′- and 3′UTRs, and the resultant bioluminescence was assessed in the DC2.4 cell line at 4, 8, 24, and 72 h after transfection. The cellular data enabled us to identify the best seven combinations of 5′- and 3′UTRs, whose translational efficiency was then assessed in BALB/c mice. Two combinations of 5′- and 3′UTRs (5′Rabb-3′mtRNR-EMCV and 5′TPL-3′Biontech) led to the most pronounced increase in the luciferase amount in the in vivo experiment in mice. Subsequent analysis of the stability of the mRNA indicated that the increase in luciferase expression is explained primarily by the efficiency of translation, not by the number of RNA molecules. Altogether, these findings suggest that 5′UTR-and-3′UTR combinations 5′Rabb-3′mtRNR- EMCV and 5′TPL-3′Biontech lead to high expression of target proteins and may be considered for use in preventive and therapeutic modalities based on mRNA.

## 1. Introduction

Lately, mRNA-based modalities have become an attractive tool for therapeutic and prophylactic use, and the amount of research in this field is growing every year, especially after the first results on approved COVID-19 mRNA vaccines from Moderna (Cambridge, MA, USA) and Pfizer/BioNTech (Mainz, Germany) [1]. In comparison with recombinant-protein drugs, the development of mRNA-based therapeutics is faster and more flexible owing to the transcription of mRNA in a cell-free in vitro system [2]. Unlike the DNA-based approach, mRNA does not have to penetrate into the cell nucleus to initiate translation.

Nonetheless, not all mRNA therapeutics are highly effective. Low stability of RNA in the cell causes premature degradation of RNA, low translational efficiency, and a decrease in the level and duration of expression of the target protein [2]. One of the key roles in the stability of mRNA molecules and in their translational efficiency is played by regulatory sequences of untranslated regions (5′UTR and 3′UTR) [3,4].

In eukaryotes, the average length of a 5′UTR varies from 50 to 200 nucleotides, but in higher eukaryotes, its length can range from several nucleotides to a thousand, indicating the existence of subpopulations of mRNAs that are subject to specific regulation [5]. It is believed that 5′UTRs of actively translated mRNAs are shorter on average [6]. Active research into properties of the 5′UTRs affecting translation has revealed some patterns. Strong secondary structure of a 5′UTR with a high GC content is associated with less efficient translation [4,7]. Nonetheless, in addition to the negative effects on translation, RNA secondary structures can give rise to higher-order interactions that are conducive to assembly of intermolecular RNA complexes with an RNA-binding protein (RBP) and regulatory RNA molecules [4].

In a 5′UTR, complicated secondary structures can occur, such as a G-quadruplex and pseudoknot. G-quadruplexes are guanine-rich structures that fold into a noncanonical tetrahelical structure with potassium ions (K^+^) chelated inside and are highly stable outside cells at temperatures above the physiological [8]. Some papers have shown that G-quadruplexes can block translation both outside cells (when tested on human *NRAS* proto-oncogene mRNA) and in eukaryotic cells (when tested on the Zic1 zinc finger protein mRNA) [9]. On the other hand, the effects of G-quadruplexes in cells remain to be studied in more detail, because apparently, in many mRNAs, G-quadruplexes are in a linear form [4]. Pseudoknots are complicated intramolecular mRNA conformations that form a knot-like three-dimensional structure and contain at least two hairpins [4]. This structure has been found in the 5′UTR of human interferon gamma (IFNG) mRNA; this pseudoknot participates in a feedback process by regulating mRNA translation through protein kinase R signaling to prevent excess interferon synthesis [10,11].

In 5′UTRs, upstream AUG codons also occur, which are located upstream of the main AUG codon. It is known that the presence of an upstream AUG in a 5′UTR reduces protein synthesis according to research on the poly(A) polymerase a (*PAPOLA*) gene, whose mRNA contains two highly conserved upstream AUGs in the 5′UTR [12]. Indeed, the presence of an upstream AUG is associated with a reduction in the translation of the main protein or with the synthesis of an upstream open reading frame, as a result of which a nonfunctional protein can be synthesized.

The presence of an internal ribosome entry site (IRES) in a 5′UTR enables the initiation of cap-independent translation through the binding of 40S ribosomal subunits to mRNA [13]. The most studied IRESs are those present in the genomes of viruses that lack a cap; for the production of viral proteins, such IRESs require a special secondary structure that is capable of initiating translation without the participation of certain translation factors [14]. Nevertheless, IRESs can also be present in cellular mRNAs that contain a cap [15]. It is reported that under the conditions where cap-dependent translation is inhibited—for example, under stress, apoptosis, or mitosis—translation can be initiated via a cap-independent pathway with the participation of an IRES to maintain a suitable level of synthesis of necessary proteins [16].

In 5′UTRs, specific motifs are present too, such as the terminal oligopyrimidine motif (TOP), which is represented by a cytosine at the 5′ end followed by continuous 4–15 pyrimidines. The presence of the TOP motif in the 5′UTR of various mRNAs allows researchers to specifically activate the translation of the entire pool of such mRNAs. Activation of translation of a TOP-containing mRNA is attributed to the mTOR complex 1 (mTORC1) nutrient-sensing signaling pathway [17]. The mRNAs containing TOP sequences mainly encode translation factors and almost all ribosomal proteins; there are ~100 such mRNAs [18].

A 3′UTR participates in such processes as alternative cleavage and polyadenylation, translation, and localization of mRNA [19], affecting mRNA stability [20]. Sequences of many 3′UTRs are extremely conserved, and their length, in contrast to 5′UTRs, varies over a wider range: from several nucleotides to several thousand. The best-known regulatory elements of 3′UTRs include AU elements that form one or several AUUUA or GUUUG sequences, which are involved in mRNA destabilization [3]. Aside from affecting the decay of mRNA, AU elements are capable of inhibiting [21,22] or enhancing translation [23,24]. For example, in resting immune cells, AU elements suppress translation, but after the stimulation of T cells with lipopolysaccharide (LPS), these elements participate in a rapid launch of protein synthesis [23,24]. This observation suggests that functions of the 3′UTR are mediated not by the *cis*-elements themselves but by specific *trans*-acting factors that bind to certain *cis*-elements.

The stability of mRNA may be influenced by the presence of microRNA-binding sites in its 3′UTR. Longer 3′UTRs correlate with lower expression levels [25]. This phenomenon has been tentatively explained as follows: longer 3′UTRs are more likely to contain microRNA-binding sites that can destabilize the mRNA. By contrast, microRNAs are reported to be less sensitive to longer 3′UTRs [26]. A possible explanation is that long 3′UTRs form closed structures with which *trans*-factors cannot interact. It has been determined that microRNA-binding sites located in the middle of a 3′UTR are less likely to mediate inhibition of mRNA translation than sites located at the ends of a 3′UTR [27]. It should be noted that a substantial number of microRNA-binding sites are located in an open reading frame; there are fewer of them in the 3′UTR, and even fewer in the 5′UTR. However, the functional influence of microRNA on mRNA destabilization is primarily due to the presence of such sites in the 3′UTR [28,29].

Various RBPs also associate with 3′UTR sequences, but binding sites for some RBPs have not been identified [30]. Nonetheless, even in the case of a known sequence of binding sites, a direct relation between the presence of a landing site and RNA stability has been difficult to discern because some RBPs stabilize RNA, some destabilize it, and others can modulate RNA stability in a context-dependent manner, as is the case for PCBP2 [31,32,33]. It should also be pointed out that the impact of most RBPs on the stability and translational efficiency of mRNA is poorly investigated. Among the best-known RBPs that bind to a 3′UTR is polypyrimidine tract–binding protein (PTB), which associates with CU-rich elements, thereby forming large loops (in RNA) important for modulation of the interaction of factors necessary for splicing [3].

Of note, despite active research into the properties of UTRs, the number of studies assessing the contribution of certain UTRs to the expression of heterologous RNAs is rather small [34,35]. Among the best-known untranslated sequences that ensure high translational efficiency are globin gene sequences. In particular, sequences of human or rabbit α- and β-globin or of rabbit β-globin have been used to improve translational efficiency of heterologous mRNAs [36,37,38].

For our study, both already known sequences of globin genes (HBB and Rabb) and regulatory sequences of constitutively expressed histone gene *H4C2* (HIST1H4B) and adenoviral tripartite leader (TPL) were chosen, which have shown good performance on translational efficiency in our previous experiments (unpublished data). In addition, we used synthetic sequence Neo-2, which has previously manifested high translational efficiency too [35]. As 3′UTRs, various combinations of mtRNR1, AES, and EMCV were chosen, which individually have previously also been found to influence mRNA stability [34,39]. In addition to the choice of individual UTRs, another question is whether protein expression depends on a specific combination of a 5′UTR and 3′UTR. Although a 5′UTR and 3′UTR do not interact directly during translation, it has been demonstrated by means of p53 mRNA as an example that the 5′UTR and 3′UTR contain complementary regions that bind translation factor RPL26, which mediates translation enhancement in response to DNA damage [40]. Thus, the aim of the current study was to find optimal combinations of a 5′UTR and 3′UTR to improve protein synthesis and mRNA stability.

## 2. Materials and Methods

### 2.1. Design of the Experiment

On the basis of both literature data and our previously obtained experimental data, we chose six sequences of 5′UTRs (TPL, HBB, rabbit β-globin [Rabb], H4C2, Neo2, and the 5′UTR from vaccine mRNA-1273 from Moderna) and five sequences of 3′UTRs (mtRNR-EMCV, mtRNR-AES, mtRNR-mtRNR, and 3′UTRs from vaccines mRNA-1273 and BNT162b2 from Moderna and BioNTech). Combining the selected sequences of 5′UTRs and 3′UTRs resulted in 30 unique combinations. Accordingly, 30 in vitro–transcribed RNAs encoding firefly luciferase (FFLuc) with various combinations of the 5′- and 3′UTRs were prepared. Aside from the 30 mRNAs carrying combinations of various untranslated regions, mRNA with regulatory regions of 5′- and 3′UTRs from vaccine BNT162b2 (BioNTech) against SARS-CoV-2 was obtained as a reference mRNA. Thus, 31 types of mRNAs were generated, which were transfected into mouse DC2.4 dendritic cells, and bioluminescence was assessed at 4, 8, 24, and 72 h after the transfection (Figure 1). The cellular data allowed us to identify the best seven combinations of 5′- and 3′UTRs that led to the highest luciferase expression. The mRNAs containing the best combinations of 5′- and 3′UTRs were encapsulated into lipid nanoparticles (LNPs) to assess translational efficiency in BALB/c mice. Intravital evaluation of bioluminescence intensity was carried out at 4, 8, 24, 48, or 72 h after injection of an mRNA encapsulated into LNPs (Figure 1). Two combinations of 5′- and 3′UTRs (5′Rabb-3′mtRNR-EMCV and 5′TPL-3′Biontech) resulted in the most pronounced upregulation of the luciferase amount in the majority of time points during the in vivo experiment on BALB/c mice. To answer the question of what raises the luciferase expression, we assessed the copy number of mRNAs carrying 5′Rabb-3′mtRNR-EMCV or 5′TPL-3′Biontech at the injection site via reverse transcription followed by quantitative PCR.

### 2.2. Cloning

Cloning of constructs for subsequent in vitro transcription of RNA was performed based on a commercial vector (pSmart; Lucigen, Middleton, WI, USA). The sequence of FFLuc (1656 bp) served as a reporter gene. Various eukaryotic and viral sequences (Appendix A) were used as 5′ and 3′UTRs, which were assembled via PCR involving overlapping oligonucleotides. Insert 5′UTR-FFLuc-3′UTR was cloned into the vector via restriction sites *Eco*RI and *Bgl*II. Downstream of the 3′UTR, there was a poly(A) tail sequence consisting of 110 adenines. The resulting vectors were used to transform NEB-stable cells (New England Biolabs, Ipswich, MA, USA), which were cultured at 30 °C and 180 rpm.

### 2.3. In Vitro Transcription

This procedure was performed as described previously [41]. To linearize the plasmid, a plasmid sample was digested with the *Spe*I restriction enzyme at a unique restriction site located immediately after the poly(A) tail. We employed 500 ng of the linearized plasmid, a buffer (20 mM DTT, 2 mM spermidine, 80 mM HEPES-KOH pH 7.4, and 24 mM MgCl_2_), 500 U of T7 RNA polymerase (Biolabmix, Novosibirsk, Russia), 200 U of ribonuclease inhibitor RiboCare (Evrogen, Moscow, Russia), and 1.5 μL of an enzyme mixture from the HighYield T7 mRNA Synthesis Kit (Jena Bioscience, Jena, Germany) as a source of inorganic pyrophosphatase. The reaction mixture also contained 6 mM of cap analog ARCA (Biolabmix, Novosibirsk, Russia) and 1.5 mM of each ribonucleoside triphosphate (Biossan, Novosibirsk, Russia). The reaction was carried out for 2 h at 37 °C, after which another 3 mM of each ribonucleoside triphosphate was added to the reaction, followed by incubation for 2 more hours. DNA was digested (eliminated) with RQ1 nuclease (Promega, Madison, WI, USA), and RNA was precipitated by the addition of LiCl to a concentration of 0.32 M and the addition of EDTA (pH 8.0) to a concentration of 20 mM, with subsequent incubation on ice for an hour. After that, the solution was centrifuged for 15 min (25,000× *g*, 4 °C). The RNA precipitate was washed with 70% ethanol, dissolved in ultrapure water, and precipitated again with alcohol according to the standard procedure. The RNA concentration was determined spectrophotometrically by means of absorbance at a wavelength of 260 nm. The desired length and homogeneity of the synthesized RNA molecules were verified by capillary electrophoresis on a TapeStation instrument (Agilent, Santa Clara, CA, USA).

### 2.4. Transfection of Cells

One day before transfection, cells were seeded in a 96-well plate for adherent cell types (Corning, New York, NY, USA) at 15,000/well in 200 μL of a medium. We used the RPMI medium (PanEco, Moscow, Russia) supplemented with 10% of fetal bovine serum (NeoFroxx, Einhausen, Germany) and 2 mM glutamine (PanEco, Moscow, Russia) for DC2.4 cells. Cells were cultivated in a CO_2_ incubator (Binder, Tuttlingen, Germany) at 5% of CO_2_ and 37 °C. The TurboFect reagent (Thermo Fisher Scientific, Waltham, WA, USA) was employed for the transfection. A mixture of two RNAs (one experimental and one internal control for normalization) was added to all cells. mRNA of NanoLuc luciferase served as the mRNA for normalization of data (the internal control).

### 2.5. Bioluminescence Detection

At 4, 8, 24, 48, or 72 h after the transfection, cells were lysed, and bioluminescence was assayed. We used the reagents D-luciferin and furimazine, manufactured by Abisense (Sochi, Russia). A working solution of luciferin (42 μg/mL) was prepared according to the manufacturer’s instructions from the kit reagents; dry furimazine was dissolved in DMSO (Servicebio, Wuhan, China) at 5 μg/mL. Lysis buffer was prepared next. Per well, 2.5 µL of the 5 µg/mL furimazine solution, 97.5 µL of 20% DMSO (Servicebio, Wuhan, China) in DPBS (Servicebio, Wuhan, China), 67 µL of DPBS, and 33 µL of the D-luciferin working solution were used; 200 μL of lysis buffer was introduced into each well of the plate and incubated for 3 min at room temperature with stirring on a shaker (Biosan, Riga, Latvia) at 300 rpm. Then, 170 μL was transferred from each well into a white microtiter plate (Sovtech, Novosibirsk, Russia). Luminescence was immediately measured on a CLARIOstar Plus device at wavelengths of 629–100 nm (gain = 3600) for luciferase and 460–60 nm (gain = 2370) for furimazine; the focal height was 8 mm. To obtain the final value, the bioluminescence of the experimental FFLuc mRNA was normalized to the bioluminescence of NanoLuc mRNA.

### 2.6. Encapsulation of mRNA into LNPs

This procedure was performed as described elsewhere [42]. In brief, a 0.2 mg/mL aqueous solution (10 mM citrate buffer, pH 3.0) of mRNA was mixed with an alcoholic solution of a lipid mixture in a microfluidic cartridge on a NanoAssemblr™ Benchtop instrument (Precision Nanosystems, Vancouver, Canada). The lipid mixture contained the following components: ionizable lipidoid ALC-0315 (BroadPharm, San Diego, CA, USA), distearoylphosphatidylcholine (Avanti Polar Lipids, Alabaster, AL, USA), cholesterol (Sigma-Aldrich, St. Louis, MO, USA), and DMG-PEG-2000 (BroadPharm, San Diego, CA, USA) at a molar ratio (%) of 46.3:9.4:42.7:1.6. The weight proportion of mRNA in the LNPs was 0.04 wt%. To form particles, the aqueous and alcoholic phases were mixed in a 3:1 ratio (*v/v*) at a total mixing rate of 10 mL/min.

After that, under sterile conditions, the particles were passed through a filter based on a 0.22 μm PES membrane (Merck Millipore, Billerica, Massachusetts, USA) and were stored at 4 °C. Next, the quality of the resulting particles was examined with the help of two parameters: particle size (Zetasizer Nano ZSP, Malvern Panalitycal, Malvern, UK) and mRNA loading. Concentration of the mRNA loaded into LNPs was determined as the difference in fluorescent signals as a result of staining (with the RiboGreen reagent; Thermo Fischer Scientific, Waltham WA, USA) of a suspension of the particles before and after their disruption. The particles were destroyed by means of a detergent, Triton X-100 (Sigma-Aldrich, St. Louis, MO, USA).

Particle size proved to be from 80 to 90 nm, and the polydispersity index was less than 0.15.

### 2.7. In Vivo Imaging of Bioluminescence

Sixty-four adult BALB/c male mice were used in the experiment, which were acquired from the Rappolovo Laboratory Animal Breeding Center (affiliated with the Russian Academy of Medical Sciences) and were maintained under conventional conditions at the Center for Experimental Pharmacology of St. Petersburg Chemical Pharmaceutical University. The animals were subdivided into eight groups via randomization, at 8 mice per group. The mice were injected with 5 μg of a test mRNA encapsulated into LNPs. The injection was intramuscular, into the right lateral large muscle of the thigh (*vastus lateralis*). After the administration of the nanoparticle suspension, the mice were kept under conventional conditions, and after 4, 8, 24, 48, or 72 h, they received inhalation anesthesia for 5 min [2.0% isoflurane (Laboratories Karizoo, S.A., Barcelona, Spain) mixed with oxygen] and an intraperitoneal injection of a D-luciferin solution (Goldbio, St. Louis, MO, USA; at 15 mg/mL, 10 µL/[g of body weight]). At 15 min after the substrate injection, a luminescence analysis was performed on an IVIS Lumina Series III instrument (PerkinElmer, Santa Clara, CA, USA). Four animals from each group were killed using cervical dislocation after 24 h; the remaining mice were killed after the last bioluminescence measurement (72 h).

### 2.8. RNA Extraction and Quantitative PCR Analysis

At 24 and 72 h after the intramuscular injections, the mice were euthanized, the thigh muscle (*vastus lateralis*) was excised, and the samples were stored at –80 °C until RNA extraction. Extraction of total RNA was performed via the phenol–chloroform method using the ExtractRNA reagent (Evrogen, Moscow, Russia). The concentration and quality of the isolated RNA were determined on a NanoDrop OneC spectrophotometer at 260 nm, after which the samples were diluted to a concentration of 1000 ng/μL.

For the reverse-transcription reaction, 1000 ng of RNA was used together with the OT-M-MuLV-RH reverse transcription kit (Biolabmix; Novosibirsk, Russia) and random hexanucleotide primers. The resulting cDNA was used to determine the number of copies of FFLuc mRNA in each sample. The assay chosen for this work was droplet digital PCR (ddPCR) with the ddPCR Supermix for Probes Kit (Bio-Rad, Pleasanton, CA, USA) in accordance with the manufacturer’s instructions.

The following sequences of primers and fluorescent TaqMan probes were used (forward: 5′-GCTACCAGGTAGCCCCAGCC-3′; reverse: 5′-GGTTTTACCGTGTTCCAGCACG-3′; probe: 5′-FAM-AGAGCATCCTGCTGCAACACC-BHQ1-3′). The 2× Supermix (Bio-Rad) and the primers and the probe ended at a final concentration of 900 and 250 nM, respectively, and 1 μL of cDNA was introduced into the reaction mixture. The final volume of the mixture was adjusted to 20 μL with ultrapure water. By means of mineral oil and a QX200 droplet generator (Bio-Rad), 20 μL of each reaction mixture was converted into droplets. Analysis of the ddPCR data was performed using Quanta Soft v.1.7.4.0917 software (Bio-Rad) to calculate the copy number of FFLuc mRNA in the samples.

### 2.9. Statistical Analysis

Statistical processing of the data obtained from the cell culture was carried out via analysis of variance (ANOVA; data were found to be normally distributed) and Bonferroni’s test as a post hoc analysis. The data obtained in the in vivo experiment were not normally distributed; therefore, the Kruskal–Wallis test and Dunn’s multiple-comparison test were performed as the post hoc analysis. Differences between groups were considered statistically significant at *p* < 0.05. The statistical analysis was performed in GraphPad Prism 9.0 (GraphPad Software, USA).

## 3. Results

In the DC2.4 cell line, we compared translational efficiency of the 31 types of luciferase-encoding mRNAs containing different combinations of UTRs. A one-way ANOVA showed an effect of the UTRs’ sequence on luciferase bioluminescence intensity in mouse dendritic cell line DC2.4 at 4, 8, 24, and 72 h after transfection [ANOVA, 4 h: F(30, 124) = 4.105, *p* < 0.001; 8 h: F(30, 124) = 6.51, *p* < 0.001; 24 h: F(30, 124) = 5.70, *p* < 0.001; 72 h: F(30, 124) = 4.951, *p* < 0.001]. In pairwise comparisons, we examined only differences from the control constructs: 5′Biontech-3′Biontech and 5′Moderna-3′Moderna. UTR combinations 5′TPL-3′mtRNR-AES and 5′Rabb-3′mtRNR-AES within FFLuc mRNA resulted in higher levels of bioluminescence as compared to 5′Moderna-3′Moderna and 5′Biontech-3′Biontech at three time points (8, 24, and 72 h after transfection). Transfection of mRNA containing 5′TPL-3′Biontech, 5′H4C2-3′mtRNR-mtRNR, or 5′Neo2-3′mtRNR-mtRNR resulted in enhanced luciferase bioluminescence after 8 and/or 24 h as compared to the mRNA carrying 5′Biontech-3′Biontech (Figure 2). Nonetheless, at 72 h after the transfection, these differences disappeared. On the contrary, UTR combination 5′H4C2-3′Biontech within mRNA did not lead to a significant increase in translational efficiency of mRNA at the early time points but caused luciferase overexpression at 72 h after the transfection (*p* < 0.05) as compared to the mRNA containing 5′Biontech-3′Biontech. Next, for further analysis of the influence of the regulatory sequences on the magnitude of luciferase expression in an in vivo experiment, six combinations of UTRs were chosen that at least manifested significant differences at one time point. In addition, combination 5′Rabb-3′mtRNR-EMCV was selected for the next stage of testing; this combination within mRNA resulted in luciferase upregulation by 18% (*p* = 0.31) at 72 h after transfection of DC2.4 cells.

In the in vivo experiment on BALB/c mice, the intensity of bioluminescence was assessed for eight combinations of 5′- and 3′UTRs within mRNA (seven experimental mRNAs and one as a reference) encapsulated into LNPs. The Kruskal–Wallis test uncovered an effect of the combinations of UTRs on luciferase bioluminescence intensity in mice at 4, 8, 24, and 48 h after the injection [Kruskal–Wallis test, 4 h: H(7, 31) = 20, *p* = 0.0056; 8 h: H(7, 30) = 19.66, *p* = 0.064; 24 h: H(7, 71) = 14.89, *p* = 0.038; 48 h: H(7, 29) = 16.15, *p* = 0.024; Figure 3]. Similar to the data obtained in the cells, there were no significant differences between the groups at 72 h after injection [72 h: Kruskal–Wallis test: H(8, 29) = 4.561, *p* = 0.71]. Only two combinations of 5′- and 3′UTRs (5′Rabb-3′mtRNR-EMCV and 5′TPL-3′Biontech) out of the seven analyzed led to a significant enhancement of bioluminescence as compared to 5′Moderna-3′Moderna at three time points (8, 24, and 48 h after transfection).

Next, to determine which properties of the UTR combinations 5′Rabb-3′mtRNR-EMCV and 5′TPL-3′Biontech caused the higher level of the target protein, we evaluated the number of copies of FFLuc mRNA at the site of injection of the mRNA–LNP composite (Figure 4). At 24 h after the administration, the number of mRNA copies did not differ significantly among the groups. Nevertheless, at the site of injection of mRNA carrying 5′Rabb-3′mtRNR-EMCV, there were 53.4 ± 45.6 million copies (mean ± SEM), and at the site of injection of mRNA containing 5′TPL-3′Biontech, there were 29.7 ± 12.4 million copies, which was 1.4–2.5 times greater relative to mRNA containing 5′Moderna-3′Moderna (20.8 ± 4.1 million copies). After 72 h, all groups showed an approximately 2- to 4-fold decrease in the copy number of FFLuc mRNA. Furthermore, at the 72-h time point, the intergroup 1.4–2.5-fold differences in the copy number disappeared. The absence of significant differences in the number of RNA copies suggested that the upregulation of luciferase was apparently not related to RNA stability. On the other hand, the observed increase in the copy number of mRNA carrying 5′TPL-3′Biontech or 5′Rabb-3′mtRNR-EMCV by 1.4–2.5-fold at the injection site after 24 h allowed us to hypothesize that there was some improvement of RNA stability. Finally, we correlated the intensity of bioluminescence with the copy number of RNA and found that combinations 5′TPL-3′Biontech and 5′Rabb-3′mtRNR-EMCV provided approximately 6-fold higher bioluminescence per RNA molecule at 24 h after the injection of the mRNA construct. These differences, however, were undetectable at 72 h after the injection. Altogether, these findings indicated that the luciferase overexpression was mediated primarily by enhanced translational efficiency and to a lesser extent by increased stability of the RNA molecules.

## 4. Discussion

In our work, we compared the impact of various combinations of 5′ and 3′UTRs within the mRNA encoding FFLuc on bioluminescence intensity in vitro and in vivo. We found that FFLuc mRNA carrying UTRs 5′TPL-3′Biontech or 5′Rabb-3′mtRNR- EMCV gives the highest bioluminescence intensity after transfection of the DC2.4 dendritic cell line and in vivo in BALB/c mice in the majority of the time points under study. Subsequent evaluation of the number of mRNA copies at the injection site revealed that the luciferase overexpression is mediated primarily by improved translational efficiency and to a lesser extent by enhanced stability of the RNA molecules. Thus, we can theorize that the contribution of the 5′UTR sequence to the observed effects is more substantial than that of the 3′UTR. Nevertheless, the best influence on the level of expression of the target protein is evidently achieved only with a combined contribution of a 5′UTR and 3′UTR to mRNA stability and translational efficiency. Overall, UTR combinations 5′Rabb-3′mtRNR-EMCV and 5′TPL-3′Biontech may be considered for use in preventive and therapeutic modalities based on mRNA.

The tripartite leader (TPL) of the adenovirus is known to function as an enhancer for translation of viral late-gene mRNAs and is thought to have an ability to initiate translation of late adenoviral mRNAs in a cap-independent manner [43,44]. The 5′UTR sequence of TPL consisted of 245 nucleotides in those studies and had a complex secondary structure. This research [43,44] indicates that TPL contains an IRES due to which TPL is capable of recruiting a ribosome, independently of interactions with the 5′ 7-methyl G cap of mRNA. On the other hand, cap-independent translation initiation is activated under stress. Consequently, we suppose that our current results on TPL are not due to cap-independent translation initiation.

In contrast, the 5′UTR of rabbit β-globin (Rabb) consists of 52 nucleotides and does not contain strong secondary structures [37]. Similarly, in our work, reference types of 5′UTR sequences from vaccines mRNA-1273 and BNT162b2 (Moderna and BioNTech) had a length of 57 nucleotides (Moderna, Cambridge, MA, USA; synthetic sequence) and 54 nucleotides (BioNTech, Mainz, Germany; 97% homologous to the 5′UTR of the human α-globin gene) and did not contain strong secondary structures (Table 1). In addition to greater length, the TPL sequence has a higher GC content and highly negative folding minimal free energy (ΔG), which correlates with inhibition of translation [4,7,45,46] (Table 1). Secondary structures of 5′ UTRs can function as major regulatory tools in relation to translation [7]. A hairpin situated not far from a cap with a free energy of −30 to 50 kcal/mol may be sufficient for blocking the access of the 43S preinitiation complex (which scans the 5′UTR to reach the start codon) and for inhibition of translation [47,48]. By contrast, there are mRNAs with a long 5′UTR that has a high GC content, and yet these mRNAs are actively translated via a cap-dependent mechanism; the 5′UTR of *LINE1* mRNA is an example (900-nt 5′UTR with a 60% GC content) [8]. This phenomenon can be explained by the fact that DEAD-box RNA helicase eukaryotic initiation factor 4A (eIF4A) can bind to mRNA immediately before translation initiation and unwind such mRNA structures, thereby clearing the path for scanning [9]. Apparently, secondary structures in a 5′UTR that arise without the participation of an RBP do not have a special inhibitory effect on translation because they are unwound by eIF4A.

On the contrary, secondary structures that emerge with the participation of an RBP are indeed capable of blocking translation initiation. A classic example is the iron-responsive element (IRE), which is a small region of a loop in a 5′UTR near the cap of a subpopulation of mRNAs involved in iron homeostasis [10]. The IRE in mRNA encoding either ferritin (iron storage protein) or ferroportin (iron transporter) associates with iron regulatory protein 1 or 2 (IRP1 or IRP2) under low-iron conditions. This event prevents the 43S complex from assembling on the 5′UTR of mRNA, thereby suppressing translation [11]. On the other hand, if IRPs bind to a 3′UTR, then they enhance the stability of the mRNA [49].

In TPL, the presence of unique *cis*- and/or *trans*-regulatory elements that ensure efficient translation is indirectly confirmed by evidence obtained via assessment of predicted mean ribosomal loading (MRL) with the Optimus 5-Prime algorithm [46]. The latter is a method for predicting MRL of a transcript through the use of convolutional neural networks. In the work just cited, data from a massive parallel reporter assay served as a training set; this assay allows us to compare translational efficiency among tens of thousands of different 5′UTRs in one experiment. On the basis of 280,000 different 5′UTR sequences, the authors of that paper demonstrated that MRL values range from 2 to 11. TPL as the 5′UTR had the highest MRL (7.7), whereas MRL of the other tested UTRs was within 5.7–6.7. Therefore, these findings suggest that the TPL contains unique sequences that are associated with high ribosome loading. It should be pointed out that the authors of ref. [46] also demonstrated that MRL inversely correlates with minimal free energy values. Thus, the MRL values of TPL are not explained simply by the presence of a complicated secondary structure.

Among the 3′UTR sequences in the mRNAs that yielded the best results were mtRNR-AES (BioNTech) and mtRNR-EMCV. Orlandini von Niessen and coworkers [34] have shown that 3′UTRs containing mtRNR1 and AES in any order promote overexpression of an upstream gene. mtRNR1 is a mitochondrial noncoding 12S ribosomal RNA that is involved in the translation of 13 mitochondrial protein-coding mRNAs [50]. AES is a distinct member of the family of Groucho/Transducin-like enhancers of split genes, regulates the transcriptional activity of androgen receptors and Notch and Wnt signaling pathways, and acts as a tumor suppressor [51]. The mtRNR-AES sequence was used in the BNT162b2 vaccine from BioNTech. The algorithm via which the authors of ref. [34] chose AES and mtRNR included a cell-based selection procedure (in which a pool of stable RNAs was isolated from cells), the creation of libraries from RNAs carrying various regulatory regions, and multiround in vitro selection of heterologous RNAs. AES and mtRNR had the lowest number of predicted binding sites for microRNAs. In the control mRNA carrying the Moderna 3′UTR, they used a sequence that was 97% complementary to a regulatory region of the human hemoglobin subunit alpha 1 (*HBA1*) gene.

The sequence of the encephalomyocarditis virus (EMCV) IRES comprises strong secondary structures and has been used to enhance the expression of heterologous RNAs [39]. Moreover, translation from the EMCV IRES is strengthened by the pyrimidine tract–binding protein (PTB) [52]. The molecular mechanism by which the IRES in a 3′UTR can enhance translation is not fully understood. For cap-independent translation initiation involving the EMCV IRES, the binding of the J-K domain to eIF4G is thought to be needed; this event is necessary for the recruitment of preinitiation complex 43S. The authors of ref. [39] suppose that translation enhancement via the ribosomes recruited downstream of a gene may proceed (at least partially) without a transfer of a ribosome to the 5′ end, probably owing to direct recognition of a start codon by ribosomal complex 43S. In other words, the 43S recruited to the 3′UTR can find the start codon in a 5′-end-independent manner because of circularization of an intermediate RNA region separating the 5′UTR and 3′UTR. On the basis of these suppositions, we can conclude that the EMCV sequence within the mRNA carrying 5′-Rabb-3′mtRNR-EMCV may play a role not only in stability but also in translational efficiency of the mRNA.

## 5. Conclusions

Selecting an optimal combination of regulatory regions, such as a 5′UTR and 3′UTR, can considerably enhance the effectiveness of mRNA-based therapeutics. Our results suggest that 5′UTR-and-3′UTR combinations 5′Rabb-3′mtRNR-EMCV and 5′TPL-3′Biontech cause high expression of target proteins and may be considered for use in preventive and therapeutic modalities based on mRNA. Although the 5′UTR and 3′UTR are currently viewed as discrete regulatory units that do not interact directly with each other, accumulating evidence points to their indirect influence on each other. Improvement of sequencing methods, the advent of new approaches (such as Ribo-seq), and the accumulation of publicly available next-generation sequencing data in this field should enable a future comprehensive analysis of patterns, which will allow us to choose an optimal combination of these regulatory regions.

## Figures and Tables

**Figure 1 biomolecules-13-01677-f001:**
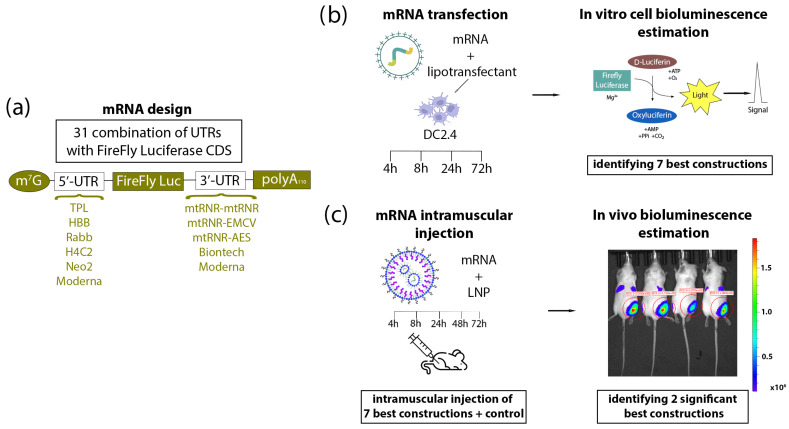
Experimental design. (**a**) Construction of mRNA. Based on previous data, six types of 5′UTR and five types of 3′UTR—presented in the figure—were chosen and combined with one another. (**b**) The DC2.4 cell line was transfected with each of the 31 various mRNAs, and bioluminescence was quantified after the addition of D-luciferin at 4, 8, 24, and 72 h after the transfection. Based on the results from the cells, seven combinations of 5′UTRs and 3′UTRs were selected that (within mRNA) caused better translation of luciferase as compared to control mRNAs containing UTRs from Moderna or BioNTech. (**c**) BALB/c mice were injected intramuscularly with one of the mRNAs (containing various combinations of UTRs) encapsulated into LNPs. Bioluminescence assessment in the mice was carried out at 4, 8, 24, 48, and 72 h after the injection.

**Figure 2 biomolecules-13-01677-f002:**
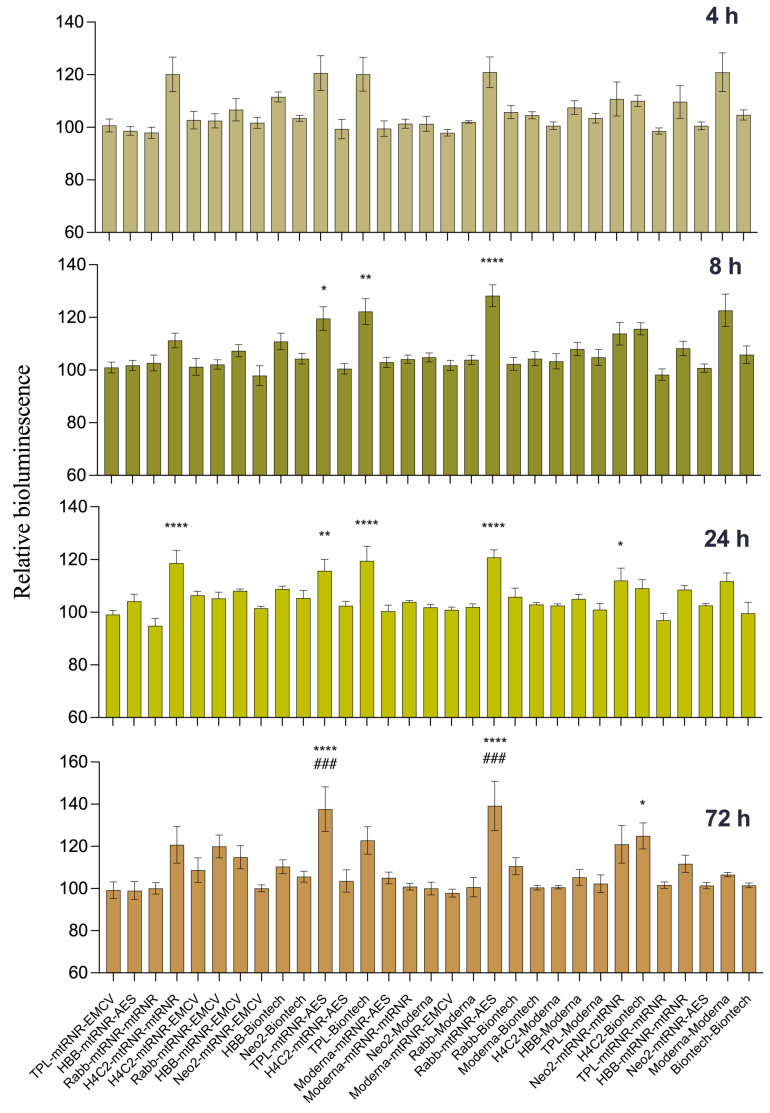
Testing of translational efficiency of mRNAs (carrying different combinations of UTRs) in DC2.4 cells. Relative intensity of luciferase bioluminescence at different time points after transfection of mRNA containing combinations of 5′- and 3′UTRs into DC2.4 cells. mRNA coding for NanoLuc luciferase served as an internal control. Data are presented as the mean ± SEM. * *p* < 0.05, ** *p* < 0.01, and **** *p* < 0.0001 as compared with 5′Biontech-3′Biontech at the corresponding time points. ^###^
*p* < 0.001 as compared to 5′Moderna-3′Moderna at respective time points.

**Figure 3 biomolecules-13-01677-f003:**
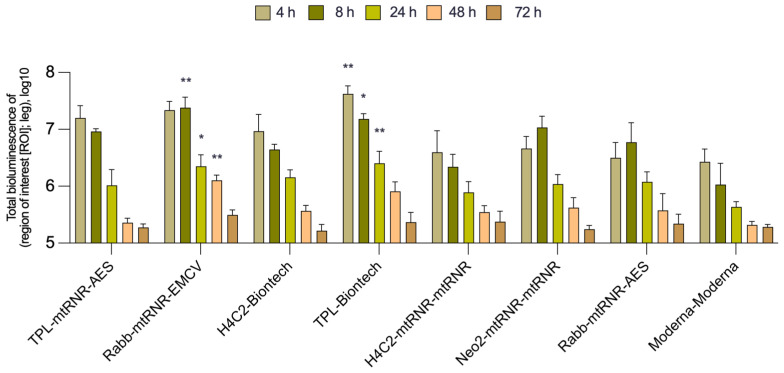
In vivo measurement of bioluminescence intensity in BALB/c mice. Data are presented as the mean ± SEM. * *p* < 0.05 and ** *p* < 0.01 as compared with 5′Moderna-3′Moderna at corresponding time points.

**Figure 4 biomolecules-13-01677-f004:**
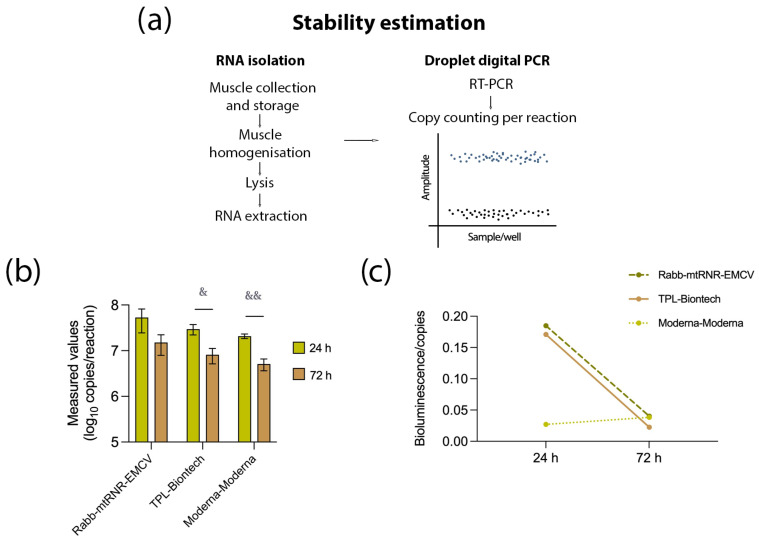
Assessment of mRNA stability. (**a**) Experimental design for the stability assay. Muscle tissue was excised at the injection site to isolate RNA. After reverse transcription, the resulting cDNA was used to quantify the target mRNA in the sample. (**b**) Determination of the number of copies of the target mRNA containing UTR combination 5′Rabb-3′mtRNR-EMCV, 5′TPL-3′Biontech, or 5′Moderna-3′Moderna at 24 and 72 h after the injection. Data are presented as the mean ± SEM. ^&^
*p* < 0.05 and ^&&^
*p* < 0.01 as compared with 5′Moderna-3′Moderna at the respective time points. (**c**) Levels of bioluminescence per copy of an RNA construct containing UTR combination 5′Rabb-3′mtRNR-EMCV, 5′TPL-3′Biontech, or 5′Moderna-3′Moderna at 24 and 72 h after the injection.

**Table 1 biomolecules-13-01677-t001:** Features of the tested 5′UTR sequences.

5′UTR	Length (Nucleotides)	GC Content	Minimal Free Energy kcal/mol	MRL [46]
TPL	245	58%	−81.9	7.7
Rabb	52	42%	−7.4	6.59
Moderna	57	47%	−4.6	5.74
BioNTech	54	50%	−9.7	6.28

## Data Availability

Raw data of this study are available upon reasonable request.

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
