# Peer review of "Effects of Combinations of Untranslated-Region Sequences on Translation of mRNA"

_biomolecules, 2023, doi:10.3390/biom13111677_

Round 1
Reviewer 1 Report
Comments and Suggestions for Authors
The effectiveness of mRNA therapy depends on the level and duration of desired protein expression, which is determined by the various cis- and trans-regulatory elements of the mRNA. In this study, the author attempts to find optimal combinations of a 5’-UTR and 3’-UTR to improve protein translation and mRNA stability. The authors found that the two sequence combinations, 5'Rabb-3'mtRNR-EMCV and 5'TPL-3'Biontech, have the best translational efficiency for the target protein and can be used as a reference for the design of mRNA therapeutic vectors.
The manuscript is well written and the illustration is presented in a good quality. The description and discussion of the results are also clear. This will provide interesting information for the reader of the journal. However, since there have been many similar studies in the past, the authors should emphasize the basis for selecting these UTR sequences for research to highlight the innovativeness of this study. In addition, due to differences in physiological regulation among species, the authors should verify the experimental results in human dendritic cells to confirm that these sequence combinations also have similar effects in human cells to expand its clinical application scope.
Comments on the Quality of English LanguageThe manuscript is well written. However, there are still some grammatical and syntax errors in the article that need to be corrected.
Author Response
Dear Editor and Referees:
Thank you for giving us the opportunity to submit a revised draft of our manuscript titled “Effects of Combinations of Untranslated-Region Sequences on Translation of mRNA” We are thankful to you for reviewing the manuscript and sharing your valuable comments and concerns with us. We have been able to incorporate most of the suggestions provided by the reviewers. We have highlighted the revisions within the manuscript.
Below, highlighted in red, are point-by-point responses to the reviewers’ comments and concerns.
Sincerely,
Vasiliy Reshetnikov
Reviewer 1.
The effectiveness of mRNA therapy depends on the level and duration of desired protein expression, which is determined by the various cis- and trans-regulatory elements of the mRNA. In this study, the author attempts to find optimal combinations of a 5’-UTR and 3’-UTR to improve protein translation and mRNA stability. The authors found that the two sequence combinations, 5'Rabb-3'mtRNR-EMCV and 5'TPL-3'Biontech, have the best translational efficiency for the target protein and can be used as a reference for the design of mRNA therapeutic vectors.
The manuscript is well written and the illustration is presented in a good quality. The description and discussion of the results are also clear. This will provide interesting information for the reader of the journal. However, since there have been many similar studies in the past, the authors should emphasize the basis for selecting these UTR sequences for research to highlight the innovativeness of this study.
Reply: Thank you for pointing this out. We have inserted into the Introduction the rationale for choosing the sequences of the UTRs used in the study.
In addition, due to differences in physiological regulation among species, the authors should verify the experimental results in human dendritic cells to confirm that these sequence combinations also have similar effects in human cells to expand its clinical application scope.
Reply: You're right, the results of this research may vary among different cell lines. These discrepancies may be caused by differences in physiological characteristics of the cell lines, in their cultivation conditions, and in specific features of translational machinery (expression of RNA-binding proteins, microRNA, lncRNA, etc.).
Sorry to disagree, but the testing on human dendritic cells is not enough to assess the possibility of clinical applications of our data. This assessment requires comprehensive testing on different cell lines, cancerous and noncancerous. In addition, functional validation of these sequences must be performed not only on reporter mRNAs but also on practically used mRNAs. Unfortunately, our work was not aimed at conducting such experiments. In our study, we focused on assessing a substantial number of UTR combinations (30) and tried to determine which factors provide improved translation.
Comments on the Quality of English Language
The manuscript is well written. However, there are still some grammatical and syntax errors in the article that need to be corrected.
Reply: Thank you for noticing. We have carefully checked all the text for grammatical and syntax errors and corrected them. If you find them again, please specify the page and line numbers where corrections need to be made.
Reviewer 2 Report
Comments and Suggestions for Authors
This manuscript investigated the effects of UTRs in mRNA translation and stability. The authors first tested the in vitro protein expression using combinations of 6 different 5’UTRs and 5 different 3’UTRs, including commonly used human gene, viral gene and therapeutic mRNA UTRs. Then the high-expression combinations were tested in vivo. Finally, two combinations were identified as the best ones. In the discussion section, the best combinations were analyzed. Besides, the experiments suggested that the high expression were achieved mainly from high translation efficiency of 5’UTRs rather than high stability of 3’ UTRs.
One main concern of the experiments is the inconsistency of in vitro results (Fig. 2) and in vivo results (Fig. 3). In this case, the claimed two highest expression combinations may not be the best in all 30 candidates.
Author Response
Dear Editor and Referees:
Thank you for giving us the opportunity to submit a revised draft of our manuscript titled “Effects of Combinations of Untranslated-Region Sequences on Translation of mRNA” We are thankful to you for reviewing the manuscript and sharing your valuable comments and concerns with us. We have been able to incorporate most of the suggestions provided by the reviewers. We have highlighted the revisions within the manuscript.
Below, highlighted in red, are point-by-point responses to the reviewers’ comments and concerns.
Sincerely,
Vasiliy Reshetnikov
Reviewer 2.
This manuscript investigated the effects of UTRs in mRNA translation and stability. The authors first tested the in vitro protein expression using combinations of 6 different 5’UTRs and 5 different 3’UTRs, including commonly used human gene, viral gene and therapeutic mRNA UTRs. Then the high-expression combinations were tested in vivo. Finally, two combinations were identified as the best ones. In the discussion section, the best combinations were analyzed. Besides, the experiments suggested that the high expression were achieved mainly from high translation efficiency of 5’UTRs rather than high stability of 3’ UTRs.
One main concern of the experiments is the inconsistency of in vitro results (Fig. 2) and in vivo results (Fig. 3). In this case, the claimed two highest expression combinations may not be the best in all 30 candidates.
Reply: Thank you for pointing this out. Indeed, the results obtained in cell lines and results obtained in mice do not match perfectly, but overall, they are largely consistent with each other. It should be noted that results in cell lines may differ from results obtained in animal models. First of all, the discrepancies may be explained by methodological specifics. Normalized data were obtained in cell lines in our study, where bioluminescence intensity in the experimental FFLuc groups was normalized to the NanoLuc luciferase signal (translated from the mRNA used for normalization, this mRNA was the same in all replicates of the assays). On the other hand, in the experiment on animals, there was no normalization, and the animals were injected only with the experimental mRNA within lipid nanoparticles. Secondly, specific features of biodistribution and the rate of RNA degradation in vitro may differ from those in vivo. Furthermore, different delivery systems were used for the cells and animals (liposomes for cells, and lipid nanoparticles for animals). Thus, the magnitude of changes may differ between cell and animal models, but it is important that the directions of changes are the same when these models are compared.
Round 2
Reviewer 1 Report
Comments and Suggestions for Authors
The authors have addressed all questions raised during the first reviewing procedure. I have found the manuscript to be much improved and recommended acceptance.